# Understanding Users' Behavior towards Applications Privacy Policies

**Salim Ullah** [1], **Muhammad Sohail Khan** [1], **Choonhwa Lee** [2] and **Muhammad Hanif** [3,*]

1    Department of Computer Software Engineering, University of Engineering and Technology, Mardan 23200, Pakistan; Engr.salim339@gmail.com (S.U.); sohail.khan@uetmardan.edu.pk (M.S.K.)
2    Department of Computer Science and Engineering, Hanyang University, Seoul 04763, Korea; lee@hanyang.ac.kr
3    Electrical and Space Engineering, Department of Computer Science, Luleå Tekniska Universitet, 97187 Lulea, Sweden
\*    Correspondence: muhammad.hanif@ltu.se

**Abstract:** Recently, smartphone usage has increased tremendously, and smartphones are being used as a requirement of daily life, equally by all age groups. Smartphone operating systems such as Android and iOS have made it possible for anyone with development skills to create apps for smartphones. This has enabled smartphone users to download and install applications from stores such as Google Play, App Store, and several other third-party sites. During installation, these applications request resource access permissions from users. The resources include hardware and software like contact, memory, location, managing phone calls, device state, messages, camera, etc. As per Google's permission policy, it is the responsibility of the user to allow or deny any permissions requested by an app. This leads to serious privacy violation issues when an app gets illegal permission granted by a user (e.g., an app might request for granted map permission and there is no need for map permission in the app, and someone can thereby access your location by this app). This study investigates the behavior of the user when it comes to safeguarding their privacy while installing apps from Google Play. In this research, first, seven different applications with irrelevant permission requests were developed and uploaded to two different Play Store accounts. The apps were live for more than 12 months and data were collected through Play Store analytics as well as the apps' policy page. The preliminary data analysis shows that only 20% of users showed concern regarding their privacy and security either through interaction with the development team through email exchange or through commenting on the platform and other means accordingly.

**Keywords:** privacy policy; android privacy; mobile application's permission; smartphone security; data confidentiality





## 1. Introduction

As we all know, smartphones have become the most essential part of our life in recent years: everyone is familiar with the usage of smartphones, and there are a lot of smartphones in the market of different operating systems, such as Android and iOS. However, there are more Android than iOS users. This is because Android is very flexible and open source, and a lot of new applications are available and developed each day in Android. People use Android smartphones for different purposes, including for communication, social media, entertainment, gaming, camera, and many other kinds of activities. Most Android users use and install third-party applications from Android Marketplace (Google Play), Amazon Appstore, and many other kinds of third-party app stores. As the Android mobile phones become widespread and powerful, many Android applications are collecting more and more sensitive data from their users through sensors, and this can be done maliciously [1].

The majority of the Android applications require permissions from users to receive access to smartphone resources when interacting with it. These permissions contain can be classified as follows. Normal permissions include battery status and internet. Alarm, time zone, vibrate, wake lock, and many more of these are permissions with low or minimum risk. Dangerous permissions include camera permission, messages read permission, record permission, storage permission, location permission accounts permission, phone gallery, etc., which fall in the dangerous category because they are riskier and can cause user data leakage if allowed. "Signature" means the Android application only gives this permission for asking permission, which is signed with a certificate. "Signature or System" means the system allows permission at the time of installation. However, these work only when the user attempts to use the application.

In this regard, the Google Play store made it compulsory for a developer to add a privacy policy in such apps that take permission from their users. The privacy policy of the app describes their usage of the apps and how the app can collect the data from the user and the data's uses. Such apps have permissions that restrict them from any malicious activities. While the application is installed, Google Play does not verify if the application is safe to use or not. Google Play relies on the bouncer, a dynamic environment to prevent itself from dangerous attacks. It cannot analyze the vulnerability of existing applications. The Google Play store thoroughly verify that the apps are safe or malicious, and according to the Google policies, it is the responsibility of the user to allow certain permissions to such applications or not. Most applications contain irrelevant permissions to the main features of apps and collect data from users that cause leaks of users' private information and can harm them, and it has been observed that most people pay no attention to permissions while installing the applications and cause very serious problems for the users.

Apps that access user data and sensitive permissions must include a privacy policy within the app and a link in the app store listing page. This protects against threats. However, the main problem is that the user does not bother to read the privacy policy of the app and understand the purpose of such sensitive permissions. The rest of the paper is organized as follows, Section 2 briefly discusses related works in the literature, Section 3 explains the proposed researched methodology, Section 4 introduces the reflective process, Section 5 introduces outcome of undertaking coursework, Section 6 provides a brief discussion and recommendations, and finally Section 7 presents conclusions and future directions. This study will help Android users follow the Google's privacy policy to read/visit the apps blog, and it will also help users to verify app permissions before installing apps to protect themselves from personal identity theft, banking and financial theft, credit card scams, and more.

## 2. Related Work

Google launched the Android operating system in 2008 for smartphones. There are many smartphone operating systems in the market, e.g., Microsoft, Symbian, and iOS, but Android brings a revolution in the smartphone industry. The demand for Android smartphones is increasing each day. After great success within a few years, Android became the number one operating system in the world, and in 2012, Google announced that it would create apps for Android phones. Developers started creating apps for Android, and the number of the Android users reached 1.9 million in 2015 [2] and 3.6 billion in 2020 [3], as shown in Figure 1.

There is no doubt that smartphones play an instrumental role in today's society. Their applications are so diverse and up to date that they have managed to make our lives drastically easier. Every government has issued advice and guidelines about social distancing and mobility restrictions. Social distancing is recommended for people in the "COVID-19" pandemic—a good example is staying home instead of going out into workplaces and public areas [4]. Many digital contact tracing apps were developed in 2020 as public health tools, which include rapid notifications, medical care, and health advice on isolation [5].

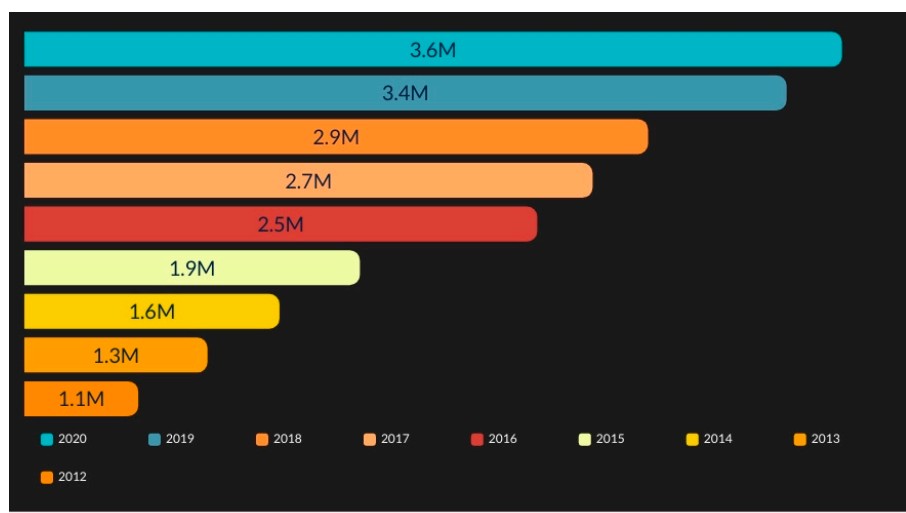

**Figure 1.** Number of smartphone users worldwide [3].

Smartphones have become part of everyone's life. Android smartphone users can install third-party applications through Android Market [6], and the majority of the apps require permission from the user to obtain access on the user device in the form of access to network, contact, device state, call logs, contacts, location, phone gallery, and camera [7]. Most modern Android smartphones implement the permission-based model to protect their users' sensitive data and prevent their privacy leakage [8].

Permissions may be required when interacting with the system API, databases, and the message-passing system. The public API describes 8648 methods, some of which are protected by permissions [9]. A marketplace such as the Google Play store thoroughly verifies if the app is legitimate or malicious. After that, it depends on the user to decide whether the applications are secure or malicious to use. Most of the applications receive irrelevant permissions from users and collect their confidential information, which could be dangerous for them.

For this purpose, the Google Play store has introduced a privacy policy regarding personal and sensitive information, and it has also introduced run time. If an app needs to use the resources or user information, it asks for proper permission from the user. The developer has to declare the permissions for the application by listing the permission in the patent file and requesting the user to approve the permissions at run time. This feature was introduced in Android operating systems 6.0 and higher. Developers are required to create a valid privacy policy when important data from the user or device are requested by the app. The inclusion of a privacy policy within the app comprehensively describes how your app can collect, use, and share data, including the types of third-party apps and with whom they will be shared. Additionally, according to Google's policies, it is the sole responsibility of the user to decide whether to allow an app asking for certain permissions on their device or not, because most of the apps on the market ask to collect data that are irrelevant to the main feature of the app, which could cause the leaking of private information or inefficient use of mobile resources.

This situation is alarming for people with low literacy who cannot decide on their own about the legality of apps. Even with the literate users of smartphones, the problem that it has been observed is that most pay no attention to the privacy permissions while installing various apps on their devices. Additionally, thus, the user becomes a victim of data theft, which may result in more serious problems for the users. For apps that request access to sensitive permissions or data, the app must link to a privacy policy on the app's store listing page, and a privacy policy should be included within the app.

Generally, it has been found in the literature that apps being uploaded to Google Play Store are not verified manually. Instead, the Google Play store relies on Bouncer, a dynamic emulation environment to protect itself from malicious app threats. It protects against

threats but cannot analyze the vulnerability of existing apps. Google also relies on the users of Android devices to allow or reject access to their device's resources in the form of permissions.

The main problem is that most users do not bother to read or understand the purpose of such permissions requested by the apps they install, allowing apps to illegally access and misuse their devices. One of the big issues is privacy concerns such as data leakage [10] and unnecessary permissions [11,12], which could lead to leakage of personal data. While some of the loopholes may be associated with liberty, portability, and ease of the system, some of the issues are due to lack of awareness and lack of technical skill of the mobile application developers [13,14]. Balebako et al. reports that most mobile app developers do not focus on privacy are unaware of any of the harm that may be caused by third-party ads and analytics tools [13].

Rashidi et al. have recently presented a comprehensive survey of security threats of mobile applications. According to their findings, as many as 70% of the applications in the Android market obtain permissions that are not even needed for the running of the applications. These unnecessary permissions are harmful in terms of extra resource usage and privacy data leakage. They have classified the threats into five broad categories: information leakage, privilege escalation, repackaging, denial of service (DoS) attack, and colluding [14].

Mobile apps have brought tremendous impact to businesses, society, and lifestyles in recent years. Various app markets offer a wide range of apps in the areas of entertainment, business, health care, and social life. Android app markets, which share the largest user base, have gained tremendous momentum since their first launch in 2008. According to the report by Android Google Play Store, the number of apps in the store reached 2.2 million in June 2016, surpassing its major competitor, the Apple App store [15]. The rise of Android phones brought about the proliferation of Android apps, resulting in an ever-growing ecosystem of applications [16].

The exponentially increasing number of Android applications, the unofficial app developers, and the existing security vulnerabilities in the Android OS encourage malware developers to take advantage of such vulnerable OSs and apps and steal private user information, inadvertently harming the apps markets and the developers' reputation [17]. Moreover, Android OS is an open-source platform that allows the installation of third-party market apps, prompting dozens of regional and international app stores to be created, such as PandaApp [18].

## 3. The Research Method

Before describing the method, we must present the aim of the privacy policy in Android Apps. Apps that use such kinds of sensitive permission gain control of a device, steal private information from users, consume excessive battery, use telephone services to steal money from users' bank accounts, and even to turn the device into a botnet zombie [18].

There are a variety of security issues on Android phones, such as unauthorized access from one app to the others (information leakage), permission escalation, repackaging apps to inject malicious code, colluding, and Denial of Service (DoS) attacks. Android applications and other applications are not allowed to give access to the resource architecture of Android. Before installing an application, the user must give access permission to the app. When the app receives the resource of an operating system, it results in the leakage of information. In this research, it has been proven that 80 to 90 percent of users did not read the privacy of the application [19,20].

Developers upload third-party applications on a daily basis to Google Play, but Google does not have the proper mechanism to ensure that users read the privacy policy before installing the applications.

Therefore, for this research, a total of seven applications were developed for the proposed study and uploaded to two different accounts. Additionally, irrelevant permissions

to the main features of apps were added in these apps. To find the result of our target area of privacy-first, we created a private URL that was used to create a private blog; this private URL is compulsory in the app store listing for apps that have permissions. This blog describes the uses of permission in apps. A flag counter option is created in the privacy blog. When a user visits the page, its entry is counted with its location (country name). Secondly, a button is created inside our apps for the privacy policy on the main screen, which is hyperlinked to our URL privacy page. When pressing this button, the user is redirected to our privacy policy blog, and it will count the visitor. Major steps involved in the proposed study are shown in Figure 2.

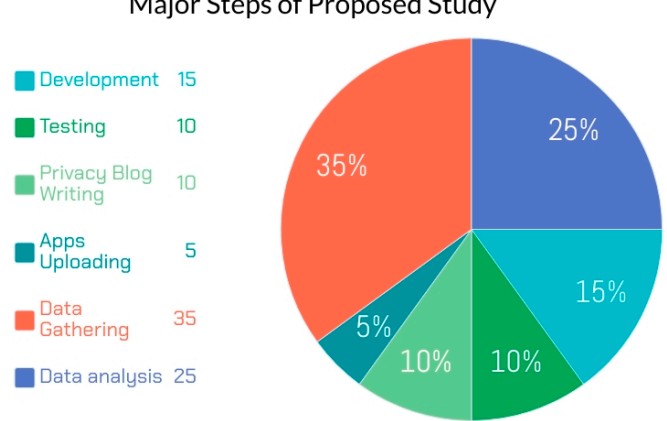

**Figure 2.** Major steps involved in proposed study.

The conceptual representation of our study is shown in Figure 3. The data gathered in this exercise were fed into an Excel sheet to perform statistical analysis on the results about app downloads. In addition, apps' download country, their population, and their literacy rates were also analyzed.

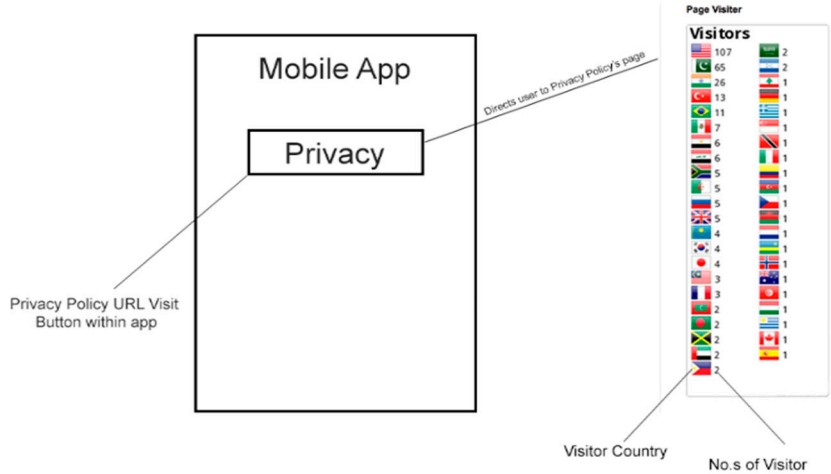

**Figure 3.** Conceptual representation of our study.

## 4. The Reflective Process

The ultimate objective of this study is to find the percentage of Android users who would allow irrelevant apps with permissions to sensitive resources on their devices and to find the percentage of Android users who would take the time to read/visit the privacy

policy blog of the apps being installed. Another objective is to obtain a general overview of global Android users' attitudes towards their devices and information privacy and correlate the collected data with the overall literacy rate country-wise. This ultimate objective may be achieved by answering the following questions.

- Will Android users provide sensitive permissions to apps and install them on their devices?
- Will Android users follow Google's privacy policy to read/visit the app's blog?
- Will Android users understand the privacy risk posed by the apps mentioned in the blog contents and be willing to withdraw from the installation of the app?

Can Google/Android App Store automatically detect the security risks posed by the apps and remove/block the apps?

Once the outcomes of the above issues have been obtained, some suggestions are provided to enhance the app's checking process at the time of uploading the app to the Google Play store.

## 5. Outcomes of Undertaking Coursework

As discussed in Section 2, a total of seven applications were developed for the proposed study; the following steps were followed for data collection and processing.

Seven applications were developed in Android Studio, and irrelevant permissions were added to each app in their manifest file that were not related to the main features of the apps. Figure 4 show one of the app's permissions. These apps were uploaded to the Google Play store, and a valid privacy policy URL was provided in the app store listing. This privacy policy URL could be visited in two ways: one was from the app's main page, in which a privacy policy button is provided to users to visit the app's privacy policy URL; the other is that users could visit the privacy policy URL from the Google Play store. The privacy policy button on the app home page was connected to our server to find the total number of privacy policy visitors coming from apps after installing the application. Below, we can see the results and the findings of the data from the last two years.

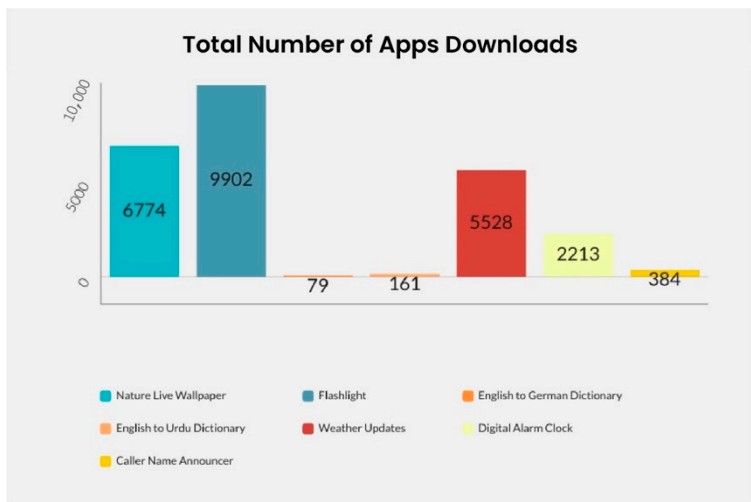

**Figure 4.** Total number of apps downloads.

As mentioned, seven very popular apps were developed from two different accounts for the study. Figures 5 and 6 show the total number of apps downloads from account 1, and Figures 7 and 8 show app downloads from account 2.

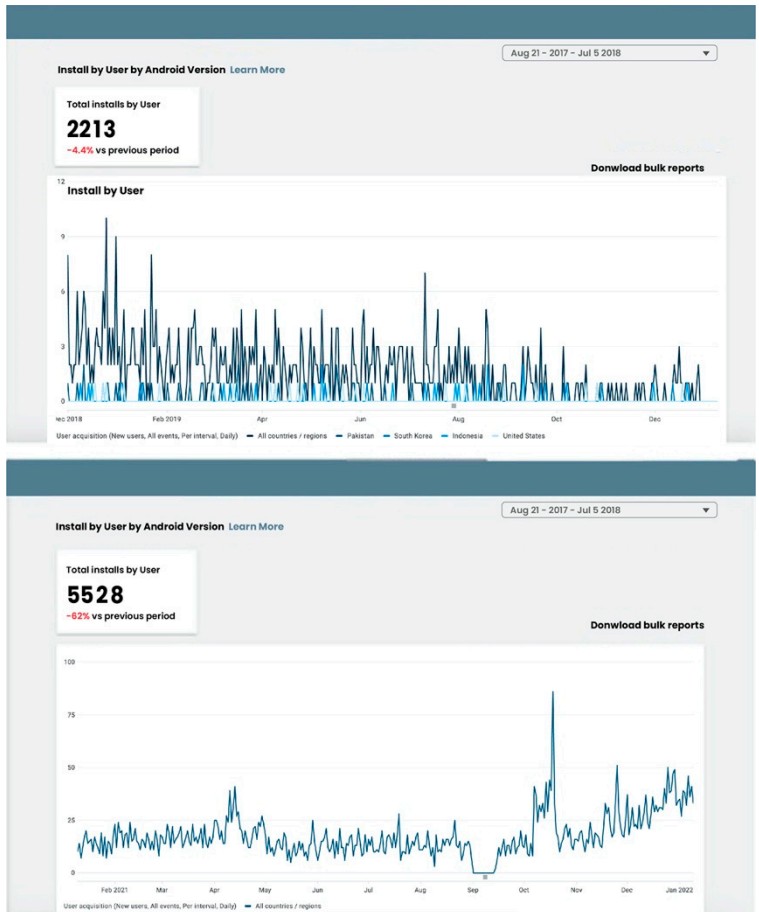

**Figure 5.** App downloads from account 1.

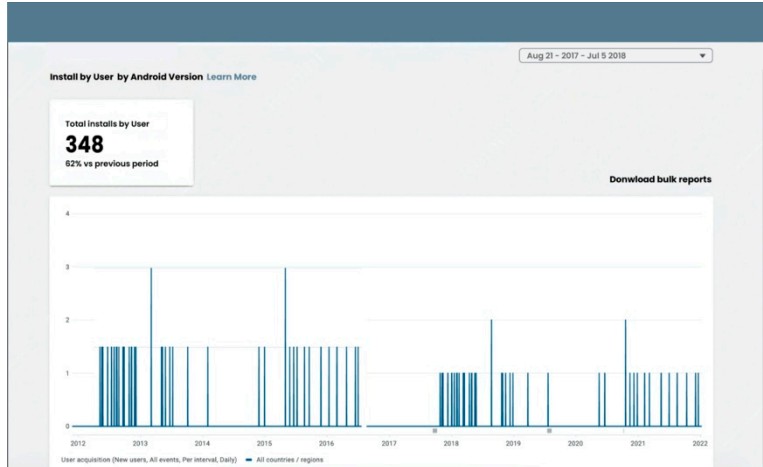

**Figure 6.** App downloads from account 1.

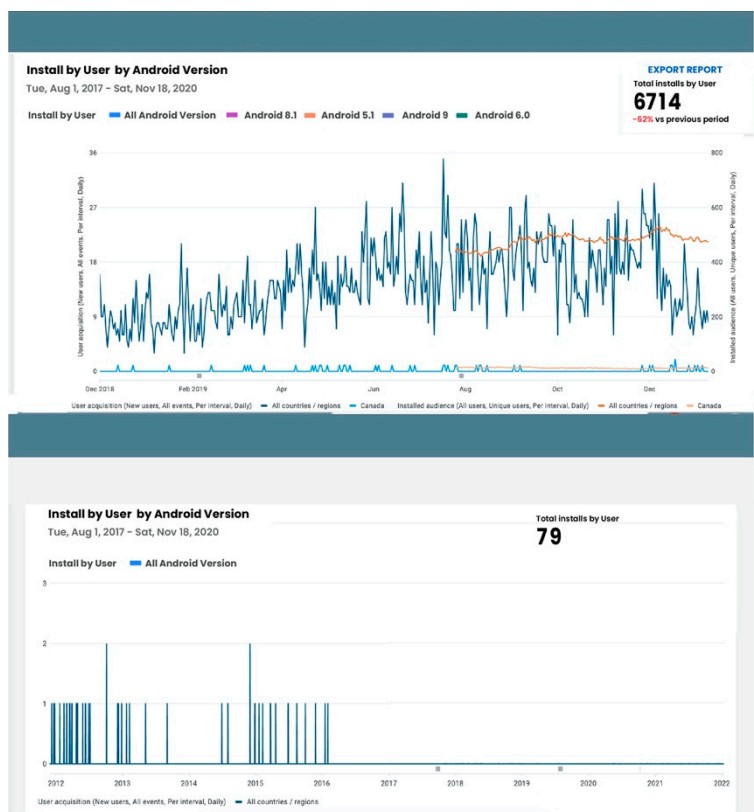

**Figure 7.** Account 2's app downloads.

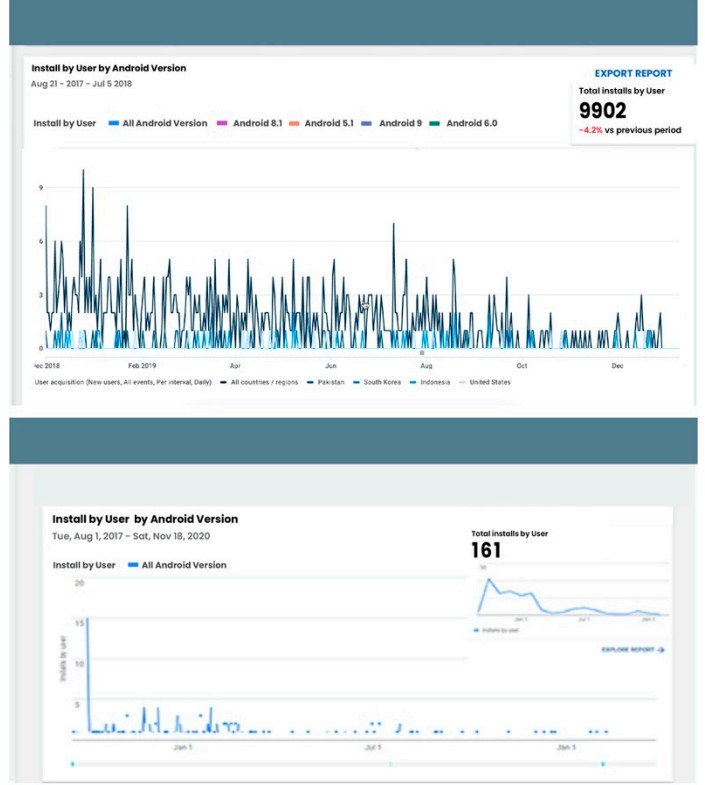

**Figure 8.** Account 2's app downloads.

Figure 9 shows the total number of users who visited the app by clicking the privacy button in the app. This picture was taken from the flag counter implemented in the blog as the % of blog visitors from each country.

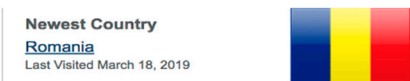

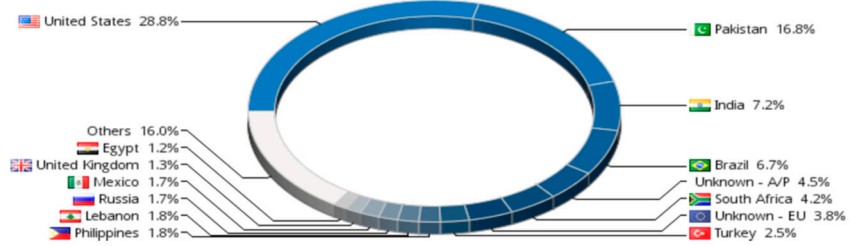

**Figure 9.** Account 1's total countries and privacy policy visitors.

## 6. Discussions and Recommendations

The seven chosen apps received a total of 25,041 app downloads and a total of 5739 privacy policy visitors. These numbers show that a much smaller number of people visit the privacy policy. Additionally, it is alarming to know that only 22% of users take privacy as a serious concern and the rest of them pay no attention, as shown in Figure 10. The privacy directly depends on the country's literacy rate, with higher literacy rates being associated with more visitors. For example, the United States' literacy rate is 86%, the total app downloads are 1232, and there were 600+ private visitors, which indicates that more than 50% of people visit the privacy policy. Similarly, if we take an example of Pakistan and India, which have lower literacy rates compared to the United States, the privacy policy visitors comprised 18% and 11%, respectively.

**Table 1.** Top countries with apps downloads with literacy rate and privacy policy visitors.

| Country Name | Policy Page Accesses | Total Download of Apps from That Country | Literacy Rate of the Country % |
|---|---|---|---|
| Brazil | 607 | 1232 | 86 |
| United State | 442 | 3837 | 69 |
| Philippines | 421 | 890 | 49 |
| Pakistan | 277 | 1482 | 55 |
| Russia | 165 | 565 | 93 |
| India | 150 | 1313 | 93 |
| South Africa | 140 | 1079 | 93 |
| Lebanon | 128 | 654 | 90 |
| Turkey | 114 | 1001 | 79 |
| Ireland | 80 | 467 | 72 |
| South Korea | 55 | 239 | 94 |
| Ukraine | 42 | 131 | 96 |
| Bangladesh | 37 | 102 | 100 |
| Greece | 36 | 89 | 93 |
| Japan | 35 | 347 | 98 |
| Iraq | 35 | 264 | 99 |
| Algeria | 35 | 263 | 95 |
| Malaysia | 33 | 186 | 99 |
| United Kingdom | 32 | 503 | 73 |
| Zimbabwe | 32 | 93 | 99 |

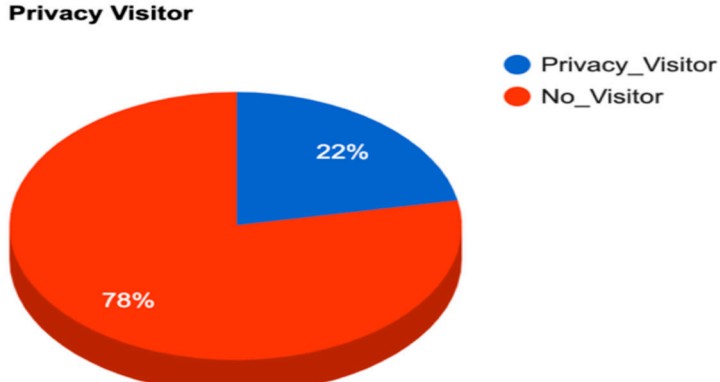

**Figure 10.** Overall privacy policy visitors. To test whether education had any relationship with how people handled their privacy, specifically the privacy of the content/data on their smartphones, the country-wise literacy rates data were obtained from [16]. These data were then augmented with information, such as country-wise Android app downloads and country-wise count of Android apps' privacy page access data from our experiments. The list was then sorted based on the number of privacy page accesses country-wise to choose the sample data. Sample data are shown in Table 1, and the full data are shown in Appendix A.

Pearson correlation between the Policy Page Accesses variable and the literacy rate of countries was calculated to find any relationship between the general education of the people of a country and their behavior toward the privacy of their Android smartphones. The Pearson correlation coefficient r = −0.58 shows a relatively strong negative correlation among the two variables, indicating that the higher the literacy rate of a country, the less its people pay attention to the privacy of the data/content on their smartphones. The outcome seems a bit strange, but a possible explanation could be that the educated people put more trust in their service providers than the less educated. It also suggests that people rely on the Google Play store's service to take care of the privacy of their data, even though Google expects them to be vigilant towards the permissions requests while installing Android applications.

Base on the presented results following two suggestions are being made:

(1) Google needs to verify applications that manually remove spam apps at the time of uploading, as the iTunes store does.
(2) A description should be added to run time permission that indicates the use of resources and needs verification.

## 7. Conclusions and Future Directions

In this paper, two different methods were used. First, we uploaded seven different apps on the Google Play store and analyzed them for 1 year. Second, we created a blog for a privacy policy. While developing the app, a "Privacy" button was created, which was a link to the blog. For these applications, we observed the number of users who visit the blog by clicking the privacy button and found that a very small number of users read the privacy blog. After obtaining this result, we concluded that there is no proper guidance about the privacy policy to the user from the Google Play store. The number of app downloads is greater more than number of privacy policy visitors as shown in Table 1. A privacy policy is a visitor also depends upon the language in which privacy is written more languages in the future can be added. Furthermore, as a future direction, we are planning to create a way in which the user must read the privacy policy of an application while installing and interacting with the application accordingly.

**Author Contributions:** Conceptualization, S.U.; Investigation, S.U., M.S.K. and M.H.; Supervision, M.S.K. and C.L.; Writing—original draft, S.U. and M.S.K. All authors have read and agreed to the published version of the manuscript.

**Funding:** This research was funded by the Korea Meteorological Administration Research and Development Program under Grant KMI 2021-01310.

**Data Availability Statement:** Data are available within the manuscript.

**Conflicts of Interest:** The authors declare no conflict of interest.

**Appendix A**

The following table shows the information about data collected and results. The country-wise literacy rate data were obtained from [16], and population data were obtained from [17].

| Country Name | Policy Page Accesses | Total Download of Apps from That Country | Literacy Rate of the Country % | Total Population |
|---|---|---|---|---|
| United State | 607 | 1232 | 86% | 323,995,528 |
| Pakistan | 277 | 1482 | 55% | 201,995,540 |
| India | 442 | 3837 | 69% | 1,266,883,598 |
| Brazil | 128 | 654 | 90% | 5,823,665 |
| South Africa | 36 | 89 | 93% | 54,300,704 |
| Turkey | 150 | 1313 | 93% | 80,274,604 |
| Philippines | 27 | 224 | 95% | 102,624,209 |
| Lebanon | 28 | 136 | 90% | 6,237,738 |
| Russia | 35 | 263 | 95% | 142,355,415 |
| Mexico | 140 | 1079 | 93% | 123,166,749 |
| United Kingdom | 35 | 264 | 99% | 64,430,428 |
| France | 33 | 186 | 99% | 66,836,154 |
| Ireland | 15 | 40 | 99% | 4,952,473 |
| Egypt | 80 | 467 | 72% | 94,666,993 |
| Japan | 32 | 93 | 99% | 126,702,133 |
| Iraq | 114 | 1001 | 79% | 38,146,025 |
| Algeria | 32 | 503 | 73% | 40,263,711 |
| Kazakhstan | 5 | 25 | 100% | 18,360,353 |
| South Korea | 37 | 102 | 100% | 50,924,172 |
| United Arab Emirates | 16 | 101 | 90% | 5,927,482 |
| Australia | 5 | 71 | 99% | 22,992,654 |
| Dominican Republic | 3 | 255 | 90% | 10,606,865 |
| Malaysia | 25 | 172 | 93% | 30,949,962 |
| Romania | 16 | 97 | 99% | 21,599,736 |
| Poland | 8 | 108 | 100% | 38,523,261 |
| Canada | 12 | 134 | 99% | 35,362,905 |
| Singapore | 5 | 36 | 96% | 5,781,728 |
| Netherlands | 10 | 73 | 99% | 17,016,967 |
| Ukraine | 25 | 101 | 100% | 44,209,733 |
| Argentina | 35 | 347 | 98% | 43,886,748 |
| Maldives | 5 | 30 | 98% | 392,960 |
| Bangladesh | 25 | 372 | 60% | 156,186,882 |
| Jamaica | 31 | 143 | 88% | 2,970,340 |
| Saudi Arabia | 55 | 239 | 94% | 28,160,273 |
| Honduras | 14 | 60 | 85% | 8,893,259 |
| China | 2 | 7 | 95% | 1,373,541,278 |
| Israel | 1 | 49 | 98% | 8,174,527 |
| Indonesia | 165 | 565 | 93% | 258,316,051 |
| Taiwan | 27 | 68 | 98% | 23,464,787 |
| Ghana | 1 | 14 | 71% | 26,908,262 |

| Country Name | Policy Page Accesses | Total Download of Apps from That Country | Literacy Rate of the Country % | Total Population |
|---|---|---|---|---|
| Austria | 8 | 18 | 98% | 8,711,770 |
| Finland | 8 | 19 | 100% | 5,498,211 |
| New Zealand | 3 | 9 | 99% | 4,474,549 |
| Zimbabwe | 27 | 47 | 84% | 14,546,961 |
| Germany | 31 | 95 | 99% | 80,722,792 |
| Greece | 27 | 82 | 97% | 0,773,253 |
| Trinidad and Tobago | 8 | 19 | 99% | 1,220,479 |
| Italy | 25 | 90 | 99% | 62,007,540 |
| Colombia | 16 | 64 | 93% | 47,220,856 |
| Azerbaijan | 30 | 100 | 100% | 9,872,765 |
| Czech | 20 | 86 | 99% | 10,644,842 |
| Malawi | 1 | 48 | 61% | 18,570,321 |
| Rwanda | 1 | 10 | 66% | 12,988,423 |
| Norway | 4 | 5 | 100% | 5,265,158 |
| Tunisia | 22 | 189 | 79% | 11,134,588 |
| Hungary | 2 | 33 | 99% | 9,874,784 |
| Uruguay | 4 | 19 | 98% | 3,351,016 |
| Spain | 28 | 101 | 98% | 48,563,476 |
| Haiti | 421 | 890 | 49% | 10,485,800 |
| Thailand | 42 | 131 | 96% | 68,200,824 |
| Algeria | 27 | 564 | 73% | 40,263,711 |
| Vietnam | 27 | 104 | 93% | 95,261,021 |
| Tunisia | 21 | 503 | 79% | 11,134,588 |
| Libya | 18 | 84 | 90% | 6,541,948 |
| Jordan | 16 | 102 | 93% | 8,185,384 |
| Guinea | 15 | 47 | 95% | 12,093,349 |
| Cambodia | 12 | 184 | 74% | 15,957,223 |
| Oman | 12 | 98 | 87% | 3,355,262 |
| Sri Lanka | 11 | 95 | 91% | 22,235,000 |
| Laos | 11 | 34 | 84% | 7,019,073 |
| Syria | 10 | 78 | 86% | 17,185,170 |
| Panama | 10 | 24 | 94% | 3,705,246 |
| Georgia | 9 | 72 | 100% | 4,928,052 |
| Somalia | 9 | 85 | 39% | 10,817,354 |
| Palestinian Territory | 8 | 29 | 98% | 1,753,327 |
| Nigeria | 9 | 210 | 51% | 186,053,386 |
| Venezuela | 7 | 57 | 95% | 30,912,302 |
| Bulgaria | 7 | 33 | 98% | 7,144,653 |
| Burma | 7 | 49 | 89% | 56,890,418 |
| Croatia | 7 | 28 | 99% | 4,313,707 |
| Reunion | 6 | 10 | 99% | 66,836,154 |
| Belgium | 6 | 29 | 99% | 11,409,077 |
| Slovakia | 6 | 17 | 100% | 5,445,802 |
| Yemen | 6 | 80 | 68% | 27,392,779 |
| El Salvador | 6 | 42 | 84% | 6,156,670 |
| Moldova | 5 | 19 | 99% | 3,510,485 |
| Macedonia | 5 | 32 | 98% | 2,100,025 |
| Guatemala | 5 | 60 | 78% | 15,189,958 |
| French Guiana | 5 | 38 | 83% | 66,836,154 |
| Afghanistan | 5 | 70 | 32% | 33,332,025 |
| Mozambique | 5 | 101 | 51% | 25,930,150 |
| Suriname | 5 | 8 | 95% | 585,824 |
| Peru | 4 | 22 | 94% | 30,741,062 |
| Hong Kong | 4 | 14 | 99% | 7,167,403 |
| Bosnia and Herzegovina | 4 | 34 | 98% | 3,861,912 |

| Country Name | Policy Page Accesses | Total Download of Apps from That Country | Literacy Rate of the Country % | Total Population |
|---|---|---|---|---|
| Belize | 4 | 41 | 83% | 353,858 |
| Ecuador | 4 | 69 | 92% | 16,080,778 |
| Puerto Rico | 4 | 22 | 93% | 3,578,056 |
| Switzerland | 3 | 10 | 99% | 8,179,294 |
| Uganda | 3 | 51 | 73% | 38,319,241 |
| Senegal | 3 | 40 | 52% | 14,320,055 |
| Kenya | 3 | 59 | 72% | 46,790,758 |
| Angola | 3 | 55 | 71% | 20,172,332 |
| Uzbekistan | 3 | 24 | 100% | 29,473,614 |
| Latvia | 3 | 12 | 100% | 1,965,686 |
| Bahrain | 3 | 19 | 95% | 1,378,904 |
| Serbia | 3 | 95 | 98% | 7,143,921 |
| Tanzania | 2 | 47 | 68% | 52,482,726 |
| Fiji | 2 | 8 | 94% | 915,303 |
| Guyana | 2 | 11 | 85% | 735,909 |
| Mongolia | 2 | 11 | 98% | 3,031,330 |
| Burkina Faso | 2 | 82 | 29% | 19,512,533 |
| Qatar | 2 | 20 | 96% | 2,258,283 |
| Slovenia | 2 | 12 | 100% | 1,978,029 |
| Brunei | 2 | 4 | 96% | 436,620 |
| Barbados | 2 | 4 | 100% | 291,495 |
| Armenia | 1 | 14 | 100% | 3,051,250 |
| Lithuania | 1 | 14 | 100% | 2,854,235 |
| Cyprus | 1 | 11 | 99% | 1,205,575 |
| French Polynesia | 1 | 1 | 98% | 285,321 |
| Togo | 1 | 30 | 60% | 7,756,937 |
| Paraguay | 1 | 10 | 94% | 6,862,812 |
| Kyrgyzstan | 1 | 8 | 99% | 5,727,553 |
| Belarus | 1 | 15 | 100% | 9,570,376 |
| Sweden | 1 | 12 | 99% | 9,880,604 |
| Sudan | 1 | 31 | 74% | 36,729,501 |
| Mauritania | 1 | 41 | 46% | 3,677,293 |
| Kuwait | 1 | 17 | 96% | 2,832,776 |
| Portugal | 1 | 29 | 94% | 10,833,816 |
| Costa Rica | 1 | 17 | 97% | 4,872,543 |
| Unknown—European Union | 192 | N/A | NA | NA |
| Unknown—Asia/Pacific Region | 149 | N/A | NA | NA |
| Unknown—Anonymous Proxy | 1 | N/A | NA | NA |
| Montenegro | N/A | 4 | 98% | 622,303 |
| Laos | N/A | 2 | 84% | 6,758,640 |
| Estonia | N/A | 1 | 99% | 1,312,442 |
| Namibia | N/A | 3 | 90% | 2,479,713 |
| Tonga | N/A | 2 | 99% | 107,122 |
| Guam | N/A | 1 | 99% | 162,896 |
| Madagascar | N/A | 3 | 64% | 24,894,551 |
| Chile | N/A | 109 | 96% | 17,909,754 |
| Denmark | N/A | 1 | 99% | 5,707,000 |
| Benin | N/A | 5 | 38% | 10,872,298 |
| Albania | N/A | 22 | 97% | 2,926,348 |
| Andorra | N/A | 2 | 100% | 77,281 |
| Aruba | N/A | 3 | 86% | 104,822 |
| **Total** | **5739** | **25,041** | | |

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
