# Peer review of "Understanding Users’ Behavior towards Applications Privacy Policies"

_electronics, doi:10.3390/electronics11020246_

Round 1

Reviewer 1 Report

The authors focus their study on the problem of privacy violation in smart applications that are offered in play stores. Specifically, the authors have studied the behavior of the users when they are concerned about their privacy while installing the applications from Google Play.

The authors have performed the real study considering seven different applications with irrelevant permission requests and they study how the users behave when their privacy is violated. The manuscript is overall well written and easy to follow and the authors have well though out their main contributions.

The provided theoretical analysis is concrete, complete, and correct and the performed testing is thorough in order to show realistic results regarding the users privacy violation. The authors should consider the following suggestions provided by the reviewer in order to improve the scientific depth of their manuscript, as well as the quality of its presentation.

Initially, in sections 1 and 2, the authors should present the provided related work by using more summative language in order to better identify the research contributions of the related work and better identify the existing research gap that the authors try to address.

Furthermore, the authors should include an additional subsection providing a detailed discussion regarding the implementation cost of the performed applications’ testing as well as the computational complexity in order to collect and analyze the corresponding data.

In Section 2, the authors should discuss several existing approaches that deal with the users’ quality of experience when accessing online applications, such as Pouli, V., et al. "Personalized multimedia content retrieval through relevance feedback techniques for enhanced user experience." 2015 13th International Conference on Telecommunications. IEEE, 2015, where the privacy concerns are weighted with respect to the service that the users enjoy.

Furthermore, the authors should include an additional subsection discussing several other techniques that have been introduced other than the ones discussed in this paper in order to prevent the privacy leakage when accessing online applications.

Additionally, the authors should include an additional subsection providing a discussion regarding the major takeaways from this study that has been performed and highlighting the main extensions and future work.

Finally, the overall manuscript should be checked for typos, syntax, and grammar errors in order to improve the quality of his presentation.

Author Response

Please find the response letter attached.

Reviewer 2 Report

The subject addressed by the author has scientific soundness and is of great importance nowadays. The purpose of the paper is clearly stated, and the research methodology is described in a comprehensive manner. The authors clearly describe the procedure and the method used in order to conduct the study. The references are adequate and relevant for the type of subject addressed. However, there are some aspect the authors should take into account:

  • The references in the text are not ordered correctly. The references should be ordered in ascending order (example: from 1 to 19). In the text after reference [1], comes reference [6], after reference [5] comes reference [8] and so on. The correct way is mention the references in the text from [1] to [19]
  • While the references used are adequate, the authors should add within the paper, information from more articles which address the subject. Thus, the reference list should be extended
  • In the findings and recommendations section, or in a separate section entitled Discussions, the authors should discuss their findings in relation to the findings of previous studies which focused on similar subjects
  • In the Conclusions section the authors should mention: the practical and theoretical implications of the research, the limitations of the research and should provide some future research directions

All in all, I enjoyed reading the paper and I want to congratulate the authors on their sound paper.

Author Response

(The authors gave the same response as above.)

Reviewer 3 Report

The manuscript discusses the understanding of the users about android apps privacy policies. The following points are observed.

  1. The language of the manuscript is not upto the marks of a standard journal. It is suggested to get it proofread by any language expert.
  2. In abstract the Line “This leads to serious privacy violation issues when an app gets illegal permission granted by a user” . How is this possible. How can a user give illegal permission?
  3. Page 1, Line 26 “showed concern regarding t…” What kind of concerns and under which context are they being analyzed is not clear.
  4. There are too many keywords and some seems to be irrelevant also like collect user data, android app privacy policy etc. how can these be termed as keywords?
  5. Page 2 line 50 “As the permission 50 defines….” Seems to be incomplete.
  6. Paper organization is missing.
  7. I do not find any motivation for the research work. What is the purpose of conducting the research need to be clearly stated.
  8. The key contributions of the manuscript need to be highlighted.
  9. Literature review is very limited. It is suggested to include latest state-of-the-art on the subject.
  10. Page 4, Line 150 “Before describing the method, I used in conducting this research I must present the….” Use of ‘I’, ‘we” etc must be avoided in a research paper.
  11. How figure 2 can be termed as a conceptual representation of the study?
  12. How visiting the privacy blog identifies that the user is interested in apps permission security? There may be situations that the privacy blog button was pressed accidently or it was made a compulsory step before completion of installation process?
  13. How is population and literacy rate be linked with apps security and privacy concerns needs to be explained in a better way.
  14. Figure 5, what is the significance of “create a free flag counter label”??
  15. In abstract it is mentioned that 7 apps were created but in section 5 it is mentioned that “total of seven applications were selected for the proposed…” both the things are contradicting. Are these seven apps developed or being used the authors is not clear to me.
  16. The recommendations made in the study seems to be already known things to the technical experts. What new has been concluded from the study is not clear.

Author Response

(The authors gave the same response as above.)

Reviewer 4 Report

The Literature review is not very much value added. I would expect that you give some review about similar research results or at least statistics about data protection problems. You just repeat the same story a few times about increase of smartfones usage and data protection problem.

We never use form "I used ..." "I must ..." in writing scientific articles.

The Conclusions are very limited. Results are also very limited and not intitive, not explainable. Only literacy rate was analysed.

Author Response

(The authors gave the same response as above.)

Reviewer 5 Report

This study focuses on understanding the users’ behaviour towards android apps privacy policy. I think the paper fits well the scope of the journal and addresses an important subject. However, a number of revisions are required before the paper can be considered for publication. There are some weak points that have to be strengthened. Below please find more specific comments:

*The abstract could be expanded a bit. I suggest adding a sentence or two highlighting the outcomes of this work and major contributions to the state-of-the-art.

*The introduction section could benefit from more statistical information. Please provide some recent statistical information highlighting the importance of different apps. The relevant statistical information should be supported by the appropriate references.

*I see that the authors discuss the importance of smartphones and different apps for everyday life in the introduction section. One important recommendation here is to introduce a short discussion that highlights the role of smartphones and different apps under the context of COVID-19 pandemic. Smartphones and different apps have been extensively used for different purposes during the pandemic (i.e., tracing of individuals who were tested positive for COVID-19). This discussion should be supported by the recent and relevant references, including the following:

  • A survey of COVID-19 contact tracing apps. IEEE Access 2020, 8, pp.134577-134601.
  • The impact of risk perception on social distancing during the COVID-19 pandemic in China. International journal of environmental research and public health 2020, 17(17), p.6256.
  • Implementing Public Health Strategies—The Need for Educational Initiatives: A Systematic Review. International Journal of Environmental Research and Public Health 2021, 18(11), p.5888.
  • Time to evaluate COVID-19 contact-tracing apps. Nature Medicine 2021, 27(3), pp.361-362.
  • Tracking and promoting the usage of a COVID-19 contact tracing app. Nature Human Behaviour 2021, 5(2), pp.247-255.

This will certainly strengthen the introduction section.

*Figure 1 can be easily reproduced. I suggest creating an original figure rather than copying it from another source. This will make the manuscript more original.

*Section 3 could benefit from a general chart that specifically outlines the major steps of the proposed methodology.

*Section 4 seems kind of short. I suggest consolidating the content of section 4 with either section 3 or section 5.

*Figures 3 and 4 could be improved by increasing the text size. Otherwise, it is hard to visualize both figures.

*Section 6 could be more detailed. The authors should provide more insights regarding the results obtained. These insights would be critical for future readers and other relevant stakeholders.

*The conclusions section should expand on limitations of this study and future research needs. I suggest listing the bullet points.

Author Response

(The authors gave the same response as above.)

Reviewer 6 Report

The authors should better describe the dataset that was produced for their study.

A better statatistical analysis should be provided instead of simple descriptive statistics e.g. using a decision tree for classifying the people who pay attention to the privacy.

In the conclusion, the authors should mention the limitations of their study and how these could be handled in a future work.

Author Response

(The authors gave the same response as above.)

Round 2

Reviewer 1 Report

The authors have not practically address the reviewers’ comments and the provided modifications and additions to the manuscript are minimal and of the minimum effort. Thus, the reviewer provides the review again to be considered and addressed by the authors.

The authors focus their study on the problem of privacy violation in smart applications that are offered in play stores. Specifically, the authors have studied the behavior of the users when they are concerned about their privacy while installing the applications from Google Play.

The authors have performed the real study considering seven different applications with irrelevant permission requests and they study how the users behave when their privacy is violated. The manuscript is overall well written and easy to follow and the authors have well though out their main contributions.

The provided theoretical analysis is concrete, complete, and correct and the performed testing is thorough in order to show realistic results regarding the users privacy violation. The authors should consider the following suggestions provided by the reviewer in order to improve the scientific depth of their manuscript, as well as the quality of its presentation.

Initially, in sections 1 and 2, the authors should present the provided related work by using more summative language in order to better identify the research contributions of the related work and better identify the existing research gap that the authors try to address.

Furthermore, the authors should include an additional subsection providing a detailed discussion regarding the implementation cost of the performed applications’ testing as well as the computational complexity in order to collect and analyze the corresponding data.

In Section 2, the authors should discuss several existing approaches that deal with the users’ quality of experience when accessing online applications, such as Pouli, V., et al. "Personalized multimedia content retrieval through relevance feedback techniques for enhanced user experience." 2015 13th International Conference on Telecommunications. IEEE, 2015, where the privacy concerns are weighted with respect to the service that the users enjoy.

Furthermore, the authors should include an additional subsection discussing several other techniques that have been introduced other than the ones discussed in this paper in order to prevent the privacy leakage when accessing online applications.

Additionally, the authors should include an additional subsection providing a discussion regarding the major takeaways from this study that has been performed and highlighting the main extensions and future work.

Finally, the overall manuscript should be checked for typos, syntax, and grammar errors in order to improve the quality of his presentation.

Author Response

please find the file attached.

Reviewer 2 Report

The authors did well in addressing the recommendations.

Author Response

We the authors are very grateful to the anonymous reviewers for their comments, which we found helpful and enlightening.

Thank you and wishing you a happy new year and happy holidays

Reviewer 4 Report

Paper has been improved significantly.

Author Response

(The authors gave the same response as above.)

Reviewer 5 Report

The authors took seriously my previous comments and made the required revisions in the manuscript. The quality and presentation of the manuscript have been improved. Therefore, I recommend acceptance.

Author Response

(The authors gave the same response as above.)

Reviewer 6 Report

A better statistical analysis should be provided instead of simple descriptive statistics e.g. using a decision tree for classifying the people who pay attention to privacy.

There are some problems in the text e.g. in the conclusion the authors state:, "The download of an app will be 1000 but the reader of privacy will be 100."

Author Response

please find the file attached.

Round 3

Reviewer 1 Report

The comments have been addressed.

Reviewer 6 Report

The manuscript could be accepted in the current form